# Transdermal System Based on Solid Cilostazol Nanoparticles Attenuates Ischemia/Reperfusion-Induced Brain Injury in Mice

**DOI:** 10.3390/nano11041009

**Published:** 2021-04-15

**Authors:** Hiroko Otake, Mizuki Yamaguchi, Fumihiko Ogata, Saori Deguchi, Naoki Yamamoto, Hiroshi Sasaki, Naohito Kawasaki, Noriaki Nagai

**Affiliations:** 1Faculty of Pharmacy, Kindai University, 3-4-1 Kowakae, Higashi-Osaka, Osaka 577-8502, Japan; hotake@phar.kindai.ac.jp (H.O.); 2033420005s@kindai.ac.jp (M.Y.); ogata@phar.kindai.ac.jp (F.O.); 2045110002h@kindai.ac.jp (S.D.); kawasaki@phar.kindai.ac.jp (N.K.); 2Department of Ophthalmology, Kanazawa Medical University, 1-1 Daigaku, Uchinada, Kahoku, Ishikawa 920-0293, Japan; naokiy@kanazawa-med.ac.jp (N.Y.); sasaki-h@k5.dion.ne.jp (H.S.)

**Keywords:** cilostazol, nanoparticle, ischemic stroke, transdermal delivery system, endocytosis

## Abstract

Cilostazol (CIL) exerted a protective effect by promoting blood–brain barrier integrity as well as improving the status of neurological dysfunctions following cerebral ischemia/reperfusion (I/R) injury. We attempted to design a 0.5% CIL carbopol gel using solid nanoparticles (CIL-Ngel), and then investigated the relationships between energy-dependent endocytosis and the skin penetration of CIL-Ngel in this study. In addition, we evaluated whether the CIL-Ngel attenuated I/R-induced brain injury in a middle cerebral artery occlusion (MCAO)/reperfusion model mouse. The particle size of CIL was decreased using a bead mill, and the CIL particles (14.9 × 10^14^ particles/0.3 g) in the CIL-Ngel were approximately 50–180 nm. The release of CIL in the CIL-Ngel was higher than that in gel containing CIL powder (CIL-Mgel), and the CIL particles were released from the CIL-Ngel as nanoparticles. In addition, the percutaneous absorption of CIL from the CIL-Ngel was higher in comparison with that from CIL-Mgel, and clathrin-dependent endocytosis and caveolae-dependent endocytosis were related to the enhanced skin penetration of CIL-NPs. In the traditional (oral administration of CIL powder, 3 mg/kg) and transdermal administration (CIL-Ngel, 0.3 g) for 3 days (once a day), the area under the plasma CIL concentration–time curves (*AUC*) was similar, although the CIL supplied to the blood by the CIL-Ngel was more sustained than that via oral administration of CIL powder. Furthermore, the CIL-Ngel attenuated the ischemic stroke. In conclusion, we designed a gel using solid CIL-NPs, and we showed that the sustained release of CIL by CIL-Ngel provided an effective treatment for ischemic stroke in MCAO/reperfusion model mice. These findings induce the possibilities of developing novel applications of CIL solid nanoparticles.

## 1. Introduction

An ischemic stroke represents the onset of a high-mortality disease and enhances social burden and causes economic disability for countries and communities. In the clinic, therapy for an ischemic stroke involves only an injection of tissue plasminogen activator (tPA), whereby the tPA dissolves any clots and restores blood supply [1]. On the other hand, the restoration of cerebral blood supply by tPA causes neuroinflammation and oxidative stress [2,3], and previous studies have reported that cerebral ischemia/reperfusion (I/R) induced excessive production of aggregating and misfolded proteins [4,5]. Thus, reperfusion in the brain exacerbates ischemic injury with neurological dysfunctions [6,7]. However, effective methods for the prevention of cerebral I/R injury have not been established in the clinic.

In a previous study, Jin et al. reported that the inhibition of inflammation-related signaling pathways could reduce I/R injury and increase functional outcome in preclinical studies of ischemic stroke [8]. Moreover, Shinohara et al. found that the administration of cilostazol (CIL) was more effective in the prevention of the recurrence of ischemic stroke, and its long-term safety and effectiveness were better than that of aspirin in patients [9]. In addition, previous reports have shown that CIL exerted a protective effect by promoting blood–brain barrier integrity [10,11] as well as improved the status of neurological dysfunctions following cerebral I/R injury [12]. We also reported that the oral or intravenous administration of CIL solid nanoparticles (CIL-NPs) prevented I/R-induced brain injury in a middle cerebral artery occlusion (MCAO)/reperfusion model mouse [13,14]. However, the administration method of CIL is limited in patients with a disturbed consciousness, whereby medical staff, such as doctors, need to inject it intravenously. Therefore, this requires the design of effective and safe formulations, allowing an evaluation of their therapeutic effect for ischemic stroke.

Application via the skin can be implemented for patients who have swallowing problems or are unconscious, and this method of administration is simple and convenient. Although drug administration through the skin permits sustained delivery [15,16], the main challenge associated with transdermal drug delivery (TDD) systems is low drug absorption, since the stratum corneum functions as a barrier. Therefore, the improvement of skin permeation using physical or chemical methods is necessary for percutaneous absorption [17]. Decades ago, it was thought that nanoparticles (NPs) could not penetrate intact skin. On the other hand, it was recently reported that NPs permeate deep into the skin depending on the type of material, surface charge, and size of the NPs [18]. Thus, nanotechnology, involving nanocarriers, nanocrystals, liposomes, polymeric NPs, solid NPs, nanoemulsions, and dendrimers, is rapidly evolving for TDD [18,19,20,21]. Our previous reports also showed that gels containing NPs of ketoprofen, indomethacin, and raloxifene showed high skin penetration via energy-dependent endocytosis, such as clathrin-dependent endocytosis (CME), caveolae-dependent endocytosis (CavME), and macropinocytosis [19,20,21]. For these reasons, TDD using solid NPs could address the abovementioned issues. Taken together, we attempted to design a gel using solid CIL-NPs (CIL-Ngel), and then investigated the relationships between energy-dependent endocytosis and skin penetration of the CIL-Ngel in this study. In addition, we evaluated whether the CIL-Ngel attenuated I/R-induced brain injury in an MCAO/reperfusion mouse model.

## 2. Materials and Methods

### 2.1. Animals

Wistar rats and Institute of Cancer Research (ICR) mice were used in this study. The 7 week old Wistar rats were purchased from Kiwa Laboratory Animals Co., Ltd. (Wakayama, Japan), and the 5 week old ICR mice (male) were obtained from Shimizu Laboratory Supplies Co., Ltd. (Kyoto, Japan). These animals were allowed free access to a CE-2 commercial diet (Clea Japan Inc., Tokyo, Japan) and water, and they were housed under standard conditions. All experiments using animals were performed in accordance with the guidelines of Kindai University, and the procedures described herein were approved on 1 April 2013 (project identification code KAPS-25-001) and 1 April 2019 (project identification code KAPS-31-010) by Kindai University.

### 2.2. Chemicals

CIL powder (CIL-MPs) was provided by Otsuka Pharmaceutical Co., Ltd. (Tokushima, Japan), and carboxypolymethylene (carbopol, Carbopol^®^ 934) was purchased from Serva (Heidelberg, Germany). 2-Hydroxyproplyl-β-cyclodextrin (HPβCD) and methylcellulose (MC) were kindly donated by Nihon Shokuhin Kako Co., Ltd. (Tokyo, Japan) and Nihon Shokuhin Kako Co., Ltd. (Tokyo, Japan), respectively. Nystatin (CavME inhibitor) and docusate sodium salt were obtained from Sigma-Aldrich Japan (Tokyo, Japan). An MFTM membrane filter (450 nm pore size) was purchased from Merck Millipore (Tokyo, Japan). Dynasore (CME inhibitor) and rottlerin (micropinocytosis inhibitor) were provided by Nacalai Tesque (Kyoto, Japan), and l-menthol, indomethacin, dimethyl sulfoxide (DMSO), isoflurane, 2,3,5-triphenyl tetrazolium chloride (TTC), and cytochalasin D (phagocytosis inhibitor) were purchased from Wako Pure Chemical Industries, Ltd. (Osaka, Japan). All other chemicals used were of the highest purity commercially available.

### 2.3. Preparation of Gel Based on CIL-NPs

CIL powder and MC were mixed and then milled using an agate mortar at 4 °C for 1 h. The milled powder (0.5 g CIL and 0.5 g MC) was suspended in a 100 mL solution containing 0.005 g of docusate sodium salt and 5 g of HPβCD, and the dispersion was stirred in a 1.5 mL tube with zirconia beads (diameter, 0.1 mm). After that, the mixture was crushed using the Micro Smash MS-100R at 5500 rpm for 30 s for a total of 30 times at 4 °C (TOMY Digital Biology Co., LTD., Tokyo, Japan). Furthermore, the dispersion containing milled CIL (CIL-NPs) was incorporated into the carbopol gel with l-menthol, and the gel containing CIL-NPs was designated CIL-Ngel. On the other hand, the gel containing CIL-MPs (CIL-Mgel) was prepared by incorporating the CIL powder and additives (MC and HPβCD) into Carbopol^®^ 934 gel with l-menthol. In this study, the compositions of CIL-Mgel and CIL-Ngel were as follows: 0.5% CIL, 0.5% MC, 0.005% docusate sodium salt, 5% HPβCD, 2% l-menthol, and 3% carbopol (*w*/*w*); their pH was adjusted to 6.8 [19,20,21]. The 3% carbopol gel containing 0.5% MC, 0.005% docusate sodium salt, 5% HPβCD, and 2% L-menthol was used as vehicle.

### 2.4. Measurement of CIL Concentration

The CIL content was determined using an HPLC LC-20AT system (Shimadzu Corp., Kyoto, Japan) with an Inertsil^®^ ODS-3 column (GL Science Co., Inc., Tokyo, Japan). Fifty microliters of sample and 50 µL of methanol containing 0.25 µg of indomethacin (internal standard) were mixed, and then 10 µL of the mixture was injected into the HPLC LC-20AT system. In the HPLC method, acetonitrile/methanol/water (35/15/50, *v*/*v*/*v*) at flow rate of 0.25 mL/min was used as the mobile phase, and CIL was detected at 254 nm at 35 °C [22].

### 2.5. Characteristics of CIL Gel

Soluble CIL and CIL-NPs in the gels were separated by centrifugation at 100,000× *g* using a Beckman Optima^TM^ MAX-XP Ultracentrifuge (Beckman Coulter, Osaka, Japan), and the solubility was determined on the basis of the concentration of soluble-CIL measured via the HPLC method described above. An atomic force microscopy (AFM) image of the CIL-NPs was generated using an SPM-9700 (Shimadzu Corp., Kyoto, Japan), and the particle-size distribution of CIL was measured using both a NANOSIGHT LM10 (QuantumDesign Japan, Tokyo, Japan) and an SALD-7100 (Shimadzu Corp., Kyoto, Japan). In this study, the refractive index used to analyze CIL particles was set at 1.60–0.10i in the SALD-7100. The viscosity of the CIL gel was analyzed using a Brookfield digital viscometer (Brookfield Engineering Laboratories, Inc., Middleboro, MA, USA), and the number of CIL-NPs was measured using the NANOSIGHT LM10. The viscosity in the NANOSIGHT LM10 was set to 1.27 mPa·s. In the evaluation of CIL gels, they were stored in the dark at 20 °C for 30 days, and the size, number, and shape of CIL-NPs were measured according to the above-described procedures.

### 2.6. Drug Release from CIL Gel

An O-ring flange (1.6 cm inner diameter (i.d.)) was placed on the MFTM membrane filter, along with a Franz diffusion cell. The reservoir chamber in the Franz diffusion cell was filled with 12.2 mL of 0.85% NaCl/10 mM phosphate buffer (pH 7.4), which was thermoregulated at 37 °C. In the experiment, the CIL gels (0.3 g) were applied into the O-ring flange on the MFTM membrane filter, and 100 µL of the sample was withdrawn from the reservoir chamber over time for 24 h [19,20,21]. The CIL concentration in the samples was measured using HPLC methods, and the size distribution and number of CIL-NPs in the samples were determined using the NANOSIGHT LM10.

### 2.7. Transdermal Penetration of CIL Gel

The hair on the abdominal area of the 7 week old Wistar rats was removed on the day prior to the experiment [19,20,21]. For the experiment, the rats were euthanized by injection with a lethal dose of pentobarbital. Then, the abdominal area was removed and collected before setting the sample on the Franz diffusion cell filled with 12.2 mL of 0.85% NaCl/10 mM phosphate buffer (pH 7.4). Next, 0.3 g of CIL gel was applied onto the O-ring flange (1.6 cm i.d.) on the skin at 4 °C (cold condition) and 37 °C (normal condition), followed by withdrawing 100 µL of solution from the reservoir chamber (sample) over time for 24 h. The cold condition was implemented to inhibit energy-dependent endocytosis [23]. Furthermore, energy-dependent endocytosis in the skin was also prevented by treatment with pharmacological inhibitors in 0.5% DMSO. The inhibitors of CavME (54 µM nystatin) [24], CME (40 µM dynasore) [25], micropinocytosis (2 µM rottlerin) [26], and phagocytosis (10 µM cytochalasin D) [24] were used in this study, and these inhibitors were used to pretreat the removed skin 1 h before treatment with the CIL gels at 37 °C. The CIL concentration in the samples was measured using HPLC methods, and the size distribution and number of CIL-NPs in the samples were determined using the NANOSIGHT LM10. The areas under the penetrated CIL concentration–time curves (*AUC*_0–24h_) were measured using the trapezoidal method.

### 2.8. Assay of Plasma CIL Concentrations

The hair on the abdominal area of the 7 week old Wistar rats was removed on the day prior to the experiment. For the experiment, CIL was orally (3 mg/kg) or transdermally (0.3 g CIL gel) administered to fasted Wistar rats once a day at 10:00 a.m. for 1 (single administration) or 3 days (repetitive administration). The CIL powder was used in the oral administration (traditional administration), whereas the CIL gel was applied for transdermal administration. Then, 100 µL of blood was drawn from the jugular vein after CIL administration, which was centrifuged at 20,400× *g* for 20 min (4 °C). Next, the CIL levels in the supernatant were measured using the HPLC method described above. The areas under the plasma CIL concentration–time curves (*AUC*_0–66h_) were measured using the trapezoidal method.

### 2.9. Induction of Focal Cerebral Ischemia/Reperfusion

The 5 week old ICR mice (41 mice) were anesthetized with 2.5% isoflurane from BS-400T (Brain Science Idea Co., Ltd., Osaka, Japan) at 37 °C, and a silicone-coated 8-0 nylon monofilament (Natsume Seisakusyo Co., Ltd., Tokyo, Japan) was inserted into the left middle cerebral artery, which was maintained for 6 h at 37 °C using a heating pad. Then, the middle cerebral artery blood flow was restored by withdrawing the nylon monofilament under anesthesia with isoflurane (3:00 p.m.) [27]. Subsequently, neurological functional were observed 4 h after the reperfusion (7:00 p.m.), and the 8 mice with no neurological symptoms (*n* = 1) or unconscious (*n* = 7) were excluded. After that, the 33 mice with the I/R injury induced by MCAO were separated into 5 groups (non-treated group (*n* = 5, neurological score 2.2 ± 0.2), group orally treated with vehicle (*n* = 5, neurological score 2.2 ± 0.2), group orally treated with CIL powder (*n* = 9, neurological score 2.3 ± 0.2), group transdermally treated with vehicle (*n* = 5, neurological score 2.4 ± 0.2), group transdermally treated with CIL-Ngel (*n* = 9, neurological score 2.4 ± 0.2)). The CIL was orally (3 mg/kg) or transdermally (0.3 g CIL gel) administered once a day (7:00 p.m., repetitive administration) and maintained for 66 h (the reperfusion was performed 70 h). Next, the brain was removed at 1:00 p.m. and three slices were cut out for analysis: from the bregma (2 mm, area_bregma_), 1 mm anterior to the bregma (2 mm, area_anterior_), and 1 mm posterior to the bregma (2 mm, area_posterior_), using brain matrices (Brain ScienceIdea Co., Ltd., Osaka, Japan). The noninfarct and infarct areas in the slices were dyed with 1.5% TTC for 15 min and monitored under a digital camera. The infarct area was analyzed with ImageJ, and the area (mm^2^) was expressed as the average area for each part of brain measured from both anterior and posterior sides of that part of brain. The infarct volume (mm^3^) was calculated using Equation (1) [13].
Infarct volume (mm^3^) = area_anterior_ × 2 + area_bregma_ × 2 + area_posterior_ × 2(1)

In this study, the neurological functional was determined on a four-score as follows: 0 = no neurological symptoms; 1 point = inability to completely extend the right front paw; 2 = rotating while crawling and falling to the right side; 3 = inability to walk without assistance; and 4 = unconscious. The mice scoring 0 and 4 were excluded.

### 2.10. CIL Contents in Brain

Focal cerebral I/R injury was induced by MCAO on the left side as described above. Three days after reperfusion, CIL was orally (3 mg/kg) or transdermally (0.3 g CIL gel) administered once a day (7:00 p.m.) and maintained for 3 days. Then, the mice were euthanized by injection with a lethal dose of pentobarbital at 1:00 p.m. Next, the brains were collected and sliced 1 mm anterior and 1 mm posterior to the center of the bregma using brain matrices [13]. The sliced brains were homogenized in methanol and centrifuged at 20,400× *g* for 20 min (4 °C). Lastly, the CIL levels in the supernatant were measured using the HPLC method described above.

### 2.11. Statistical Analysis

All values are expressed as the mean ± standard error (SE). The significant differences between mean values (*p* < 0.05) were analyzed according to Student’s t-test and Dunnett’s multiple comparisons.

## 3. Results

### 3.1. Development of the CIL-Ngel

Figure 1 shows the particle size of CIL in the CIL gels. The particle size of CIL in the CIL-Mgel was 65.1 ± 0.27 µm, which was shifted to the nanoscale via treatment with the bead mill method. Figure 2 shows the dispersity, solubility, and viscosity of the CIL-Ngel. The CIL particles in the gel were uniformly dispersed, with a lower SE calculated in the CIL-Ngel than in the CIL-Mgel (Figure 2A). Although, the dissolved CIL levels in the CIL-Ngel were 4.2-fold higher than in the CIL-Mgel (Figure 2B), the amount of dissolved CIL in the CIL-Ngel remained low, with 92.8% of the CIL in the NP state. On the other hand, the viscosity was similar between the CIL-Mgel and CIL-Ngel (Figure 2C). Figure 3 shows the changes in the particle size and number of CIL in the CIL-Ngel 1 month after the preparation. No difference in particle-size distribution, number, or shape of CIL-NPs was observed in the CIL-Ngel after 1 month.

### 3.2. Mechanism of CIL Transdermal Penetration in the Rat Skin Treated with the CIL-Ngel

Figure 4 shows the CIL release from the CIL gel. The release of CIL from the CIL-Ngel was significantly higher in comparison with that from the CIL-Mgel. In addition, CIL-NPs were not observed in the reservoir chamber with CIL-Mgel; however, they were detected in the reservoir chamber with CIL-Ngel. The size and number of CIL-NPs in the reservoir chamber were 50–400 nm and 1.79 ± 0.13 × 10^8^ particles/mL, respectively, 24 h after application. Figure 5A–C show the transdermal penetration of CIL in the CIL-Mgel and CIL-Ngel under the cold (4 °C) and normal conditions (37 °C). The penetration profile of CIL in the CIL-Ngel was significantly higher than that in the CIL-Mgel under the normal condition. In contrast to the result obtained for the release of CIL from the CIL-Ngel (Figure 3), no CIL-NPs were detected in the reservoir chamber after treatment with CIL-Ngel, since the number of CIL-NPs in the reservoir chamber was below the detection limit of the NANOSIGHT LM10 24 h after treatment. Furthermore, the transdermal penetration of CIL was significantly prevented under the cold condition (4 °C), and the CIL levels in the reservoir chamber were not significantly different between the CIL-Mgel and CIL-Ngel. Figure 5D,E show the effect of energy-dependent endocytosis on the transdermal penetration of CIL-Ngel using endocytosis inhibitors. Rottlerin and cytochalasin D did not affect the transdermal penetration; however, both nystatin and dynasore significantly decreased the transdermal penetration of the CIL-Ngel. In particular, dynasore strongly prevented the transdermal penetration of the CIL-Ngel, whereby the *AUC*_0–24h_ in the rat skin treated with dynasore was 41.3% that of the vehicle-treated group (control).

### 3.3. Therapeutic Effect of the CIL-Ngel on Ischemic Stroke in MCAO/Reperfusion Model

Figure 6A shows the changes in plasma CIL concentration after the oral and transdermal administration of CIL. The plasma CIL level peaked at 3 h, before gradually decreasing to a low level 24 h after the oral administration of the CIL powder. On the other hand, the plasma CIL level was increased at 3 h, as well as at 24 h, after the transdermal administration of the CIL-Ngel. Figure 6B,C show the changes in plasma CIL concentration in the rats repetitively administered CIL. The *AUC*_0–66h_ was similar for oral and transdermal administration. Figure 6D shows the changes in CIL concentration in the brain after the oral and transdermal administration of CIL. The CIL contents at the noninfarct and infarct sites of MCAO/reperfusion model mice orally treated with CIL were 0.71 and 0.30 µmol/mg protein, respectively. During the repetitive administration of CIL-Ngel for 3 days, a high level of CIL was delivered into the brain. The CIL content at the noninfarct site of the brain was 1.66 µmol/mg protein, whereas that at the infarct site of the brain was lower (0.94 µmol/mg protein) after the traditional administration of CIL-Ngel. Figure 7 shows the changes in ischemic stroke in MCAO/reperfusion model mice following the oral and transdermal administration of CIL. Ischemic stroke was not observed in the sham control mice (data not shown). Both the oral and the transdermal administration of CIL attenuated ischemic stroke; however, the protective effect of the CIL-Ngel was significantly higher than that seen with the oral administration of CIL powder. Despite the similar *AUC*_0–66h_ following the oral and transdermal administration (Figure 6C), the infarct volume in the mouse brain transdermally treated with the CIL-Ngel was 68% that of the volume in the brain orally treated with CIL powder (Figure 7D).

## 4. Discussion

CIL, 6-[4-(1-cyclohexyl-1*H*-tetrazol-5yl)butoxy]-3, 4-dihydro-2(1*H*)-quiolinone, can decrease the degree of neuronal cell death following transient cerebral ischemia [28], and it provides a neuroprotective effect against brain injury induced by I/R [29,30]. It was suggested that CIL inhibits the inflammatory reaction of microglia or tumor necrosis factor alpha after ischemic insult, resulting in a neuroprotective effect [28]. Therefore, CIL is expected to be effective for treating transient cerebral ischemia. In this study, we designed a CIL-Ngel (gel based on solid CIL-NPs) and showed that the CIL-Ngel was better for the therapy of I/R-induced brain injury.

The skin is the largest multilayered organ of the body, and TDD permits sustained delivery [15,16]. In particular, the skin route is useful for unconscious patients, and it provides assurance of more consistent serum drug levels [15,16]. On the other hand, the skin plays a role as a biologic protector, preventing the entry of foreign substances into the body; thus, improvement of the low absorption is a major challenge associated with transdermal drug administration. In order to overcome this hurdle, various techniques for cutaneous penetration enhancement using chemical, physical, and NP approaches have been proposed [18,19,20,21]. We also reported that 2% l-menthol increased the skin penetration of solid NPs, whereby indomethacin and raloxifene NPs presented high percutaneous absorption in combination with l-menthol [20,21]. Moreover, energy-dependent endocytosis is related to the enhanced skin penetration of NPs [19,20,21]. Taken together, we prepared a transdermal system using solid CIL-NPs in this study.

Firstly, we attempted to prepare a carbopol gel containing solid CIL-NPs following our previous reports [19,20,21]. The bead mill treatment decreased the particle size of CIL to 50–180 nm (Figure 1), with 93.8% of the total CIL in the gel uniformly present as solid CIL-NPs (Figure 2). In addition, the CIL particles in the gel were stable and retained their nanosize 1 month after preparation (Figure 3). Furthermore, the CIL-Ngel released CIL-NPs during the in vitro release examination using a Franz diffusion cell set in an MFTM membrane filter (Figure 4). These results validated the practicality of the CIL-Ngel.

Next, the skin penetration of the CIL-Ngel was demonstrated (Figure 5A). It was reported that NPs can permeate deep into the skin depending on the type of material, surface charge, and size of the NPs [18]. In this study, transdermal penetration was also enhanced following the preparation of NPs with respect to the CIL-Mgel. On the other hand, it is important to elucidate the mechanism underlying the high skin penetration of CIL-NPs in the CIL-Ngel. The precise mechanism for the transdermal penetration of NPs is not clearly defined; however, recent reports showed that the activity of endocytosis increases during the cellular uptake of NPs [19,20,21]. Therefore, we investigated the relationships between endocytosis and transdermal penetration in the CIL-Ngel. He et al. reported that the cold condition (4 °C) was able to inhibit the activity of energy-dependent endocytosis [23]. In this study, we demonstrated the effect of endocytosis inhibition on the transdermal penetration of CIL under the cold condition [23] using pharmacological inhibitors [24,25,26]. The inhibition of energy-dependent endocytosis under the cold condition (4 °C) inhibited the transdermal penetration of CIL (Figure 5B,C), with nystatin (CavME inhibitor) and dynasore (CME inhibitor) significantly attenuating the transdermal penetration of CIL in the CIL-Ngel (Figure 5D,E). From these results, it was suggested that both CavME and CME promoted the transdermal penetration of CIL.

Furthermore, we evaluated whether the CIL was delivered via percutaneous absorption to the infarct site of the brain in mice with I/R-induced brain injury. The retention time of plasma CIL was longer in the CIL-Ngel, and the CIL in plasma and the brain was significantly higher in comparison with the oral administration of CIL powder (56.4%) (Figure 6A,B,D). On the other hand, the CIL concentration in non-ischemia site was higher than that in ischemia site (Figure 6D). These results suggested that the blood flow may be limited under the the ischemia, resulting in the delivery of CIL in ischemia site was not lower than that of non-ischemia site. It was important to measure whether treatment with the CIL-Ngel prevented ischemic stroke. Therefore, we also investigated the therapeutic effect of the CIL-Ngel on ischemic stroke in MCAO/reperfusion model mice (Figure 7). The CIL-Ngel significantly attenuated ischemic stroke. Furthermore, *AUC*_0–66h_ was similar between orally and transdermally treated mice (Figure 6C); however, the infarct area in MCAO/reperfusion model mice treated with the CIL-Ngel was higher than that following the oral administration of CIL powder. Taken together, we hypothesize that the CIL-NPs released from the CIL-Ngel penetrated through the system via CavME and CME and dissolved in the skin, since no CIL-NPs were detected in the reservoir chamber following the treatment with CIL-Ngel in the in vitro skin penetration study using a Franz diffusion cell. The dissolved CIL was absorbed into the blood and delivered to the brain regardless of the presence of infarct. Moreover, the CIL-Ngel provided assurance of more consistent plasma CIL levels in comparison with that following the oral administration of CIL powder, resulting in a strong inhibition of the onset of ischemic stroke in MCAO/reperfusion model mice. Yoneyama et al. reported that CIL promoted neuronal repair following cerebral ischemic injury [12], while Horai et al. showed that CIL exerted a protective effect by promoting blood-brain barrier integrity [11]. Our results in this study support these previous reports, showing that the sustained release of CIL may be effective in the treatment of ischemic stroke in MCAO/reperfusion model mice.

Further studies are needed to clarify the relationships between percutaneous absorption and endocytosis pathways in the CIL-Ngel. In addition, it is important to establish effective therapy for ischemic stroke. In future work, we plan to investigate the therapeutic effect of a combination of CIL and tPA.

## 5. Conclusions

We designed a gel incorporating solid CIL-NPs, and we found that CavME and CME were responsible for the enhanced skin penetration of CIL in the CIL-Ngel. In addition, we showed that the sustained supplementation of CIL represents an effective treatment for ischemic stroke in MCAO/reperfusion model mice. These findings demonstrate the possibilities of developing novel applications using CIL-NPs.

## Figures and Tables

**Figure 1 nanomaterials-11-01009-f001:**
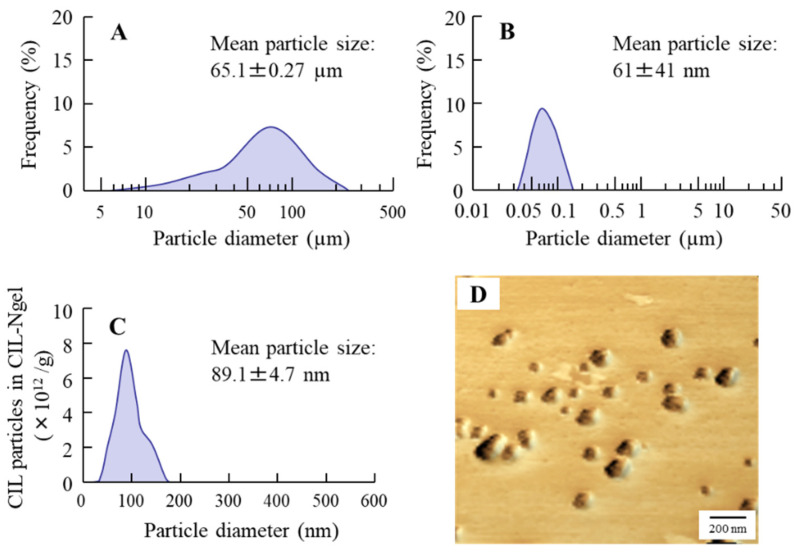
Particle-size frequencies and atomic force microscopy (AFM) image of cilostazol (CIL) in 0.5% CIL carbopol gel using solid nanoparticles (CIL-Ngel). (**A**,**B**), Particle-size frequencies of CIL in the gel containing CIL powder (CIL-Mgel) (**A**) and CIL-Ngel (**B**) using the SALD-7100. (**C**) Particle-size frequencies of CIL in the CIL-Ngel using the NanoSight LM10. (**D**) AFM image of CIL in the CIL-Ngel using the SPM-9700. The CIL particles in the CIL-Ngel were approximately 50–180 nm.

**Figure 2 nanomaterials-11-01009-f002:**
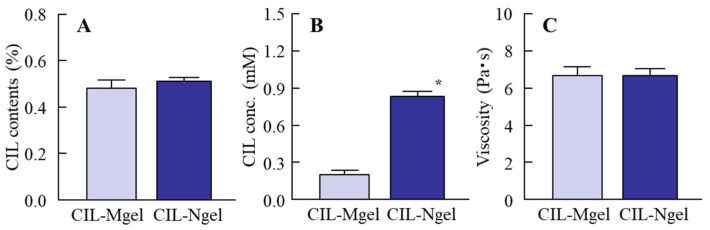
The dispersity (**A**), drug solubility (**B**), and viscosity (**C**) in the CIL-Ngel (*n* = 6); * *p* < 0.05 vs. CIL-Mgel for each category. The CIL particles in both gels were uniform. On the other hand, the dissolved CIL levels in the CIL-Ngel were significantly higher than in the CIL-Mgel.

**Figure 3 nanomaterials-11-01009-f003:**
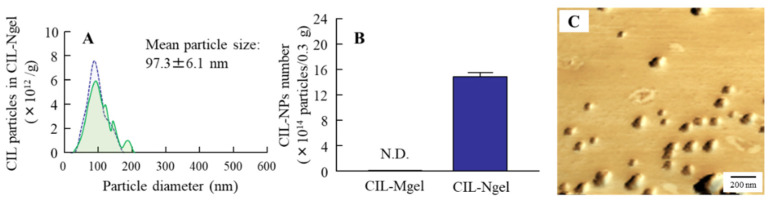
The stability of CIL particles in the CIL-Ngel. (**A**) Changes in size frequency in the CIL-Ngel immediately (blue) and 1 month (green) after preparation. (**B**) Changes in the number of CIL-NPs in the CIL-Ngel 1 month after preparation. (**C**) AFM image of CIL-NPs in the CIL-Ngel 1 month after preparation. N.D., not detectable; *n* = 5. The CIL nanoparticles (CIL-NPs) in the CIL-Ngel were stable for 1 month.

**Figure 4 nanomaterials-11-01009-f004:**
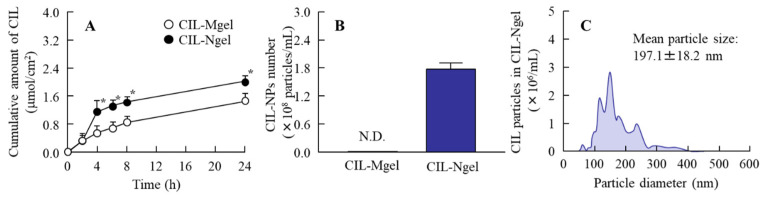
Profiles of CIL release from the gel through 450 nm pore membranes. (**A**) CIL release from the CIL-Mgel and CIL-Ngel through the membranes. (**B**,**C**) Number (**B**) and size frequencies (**C**) of CIL-NPs released from the CIL-Ngel 24 h after application (*n* = 7); * *p* < 0.05 vs. CIL-Mgel for each category. The release of CIL from the CIL-Ngel was higher than that from the CIL-Mgel, and the CIL was released from the CIL-Ngel as NPs.

**Figure 5 nanomaterials-11-01009-f005:**
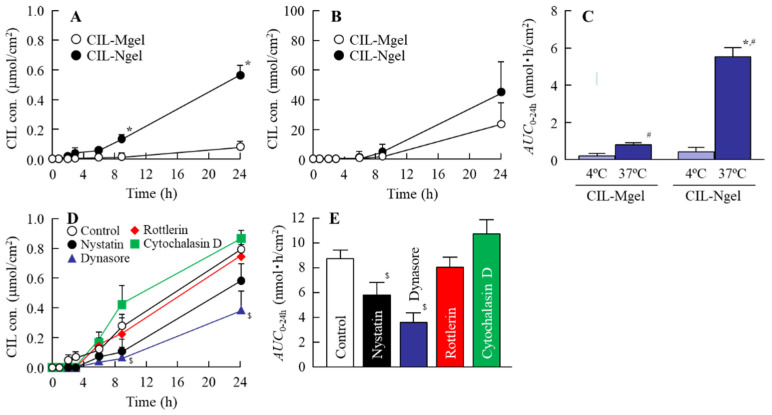
Relationships between energy-dependent endocytosis and transdermal penetration of the CIL-Ngel. (**A**) Transdermal penetration of CIL in the CIL-Mgel and CIL-Ngel under the normal condition (37 °C). (**B**) Transdermal penetration of CIL in the CIL-Mgel and CIL-Ngel under the cold condition (4 °C). (**C**) Area under the penetrated CIL concentration–time curves (*AUC*_0–24h_) of the CIL-Ngel on the skin under the normal and cold conditions. (**D**,**E**) Penetration profile (**D**) and *AUC*_0–24h_ (**E**) of the CIL-Ngel in the skin treated with endocytosis inhibitors (nystatin, dynasore, rottlerin, and cytochalasin D; *n* = 6–8); * *p* < 0.05 vs. CIL-Mgel for each category, ^#^
*p* < 0.05 vs. 4 °C treatment group for each category. ^$^
*p* < 0.05 vs. control for each category. The transdermal penetration of CIL-NPs in the CIL-Ngel was attenuated by the treatment with nystatin and dynasore.

**Figure 6 nanomaterials-11-01009-f006:**
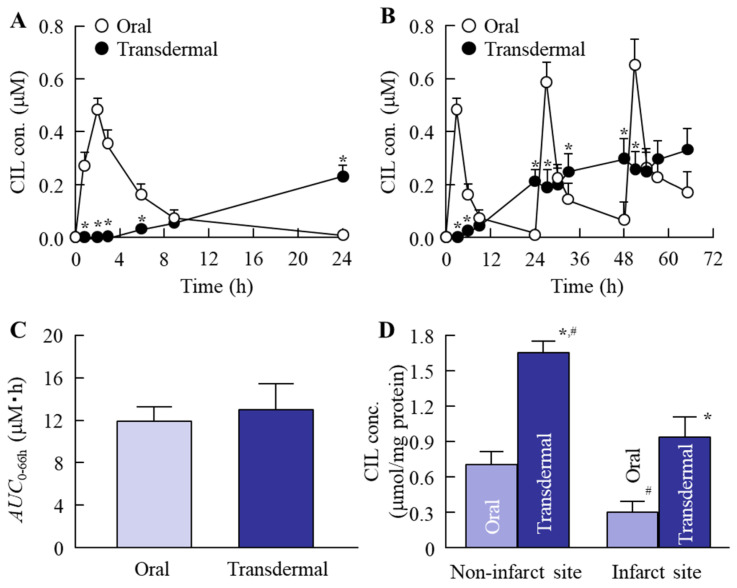
The percutaneous absorption of the CIL-Ngel. (**A**) Plasma CIL concentration in rats following a single oral or transdermal administration of CIL. (**B**,**C**) Plasma CIL concentration (**B**) and *AUC*_0–66h_ (**C**) in rats following repetitive oral or transdermal administration of CIL. (**D**) CIL concentration in brain of middle cerebral artery occlusion (MCAO)/reperfusion model mice after the oral and transdermal administration of CIL. The administration of CIL was performed once a day for 3 days in the repetitive administration. Oral, group orally treated with CIL powder; transdermal, group transdermally treated with CIL-Ngel. *n* = 5–8. * *p* < 0.05 vs. CIL orally treated group for each category, ^#^
*p* < 0.05 vs. infarct site of MCAO/reperfusion model mice transdermally treated with CIL for each category. The percutaneous absorption of CIL from the CIL-Ngel increased linearly for 24 h, and CIL was delivered to the brain following administration of the CIL-Ngel. Although the *AUC*_0–66h_ with the CIL-Ngel was similar to that following the oral administration of CIL macroparticles (CIL-MPs), the CIL content was higher than that in the mice treated with CIL powder.

**Figure 7 nanomaterials-11-01009-f007:**
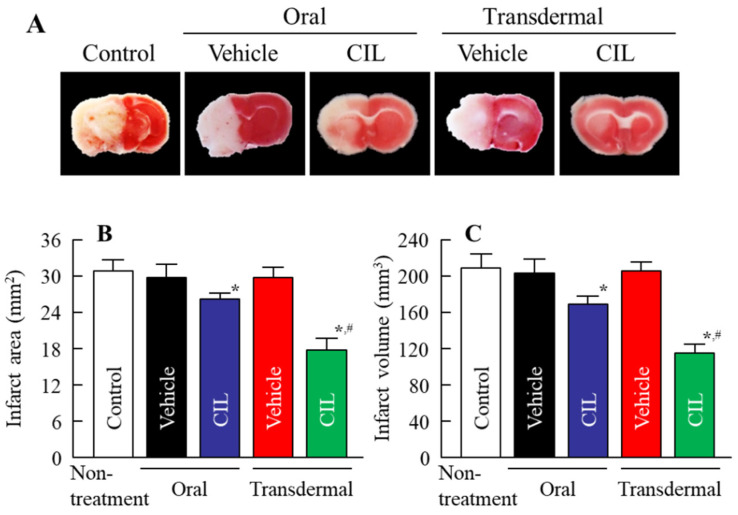
Protective effect of CIL-Ngel on ischemic stroke in MCAO/reperfusion model mice. (**A**) Image of the brain of MCAO/reperfusion model mice following the oral and transdermal administration of CIL. (**B**,**C**) Infarct area (**B**) and volume (**C**) of MCAO/reperfusion model mice following the oral and transdermal administration of CIL. The mice were separated randomly into 5 groups, and infarct area was measured by the person who was blind to the group. The CIL-Ngel was transdermally administered once a day for 3 days. *n* = 5–9. * *p* < 0.05 vs. control for each category, ^#^
*p* < 0.05 vs. CIL orally treated MCAO/reperfusion model mice for each category. The CIL-Ngel attenuated ischemic stroke, and the therapeutic effect was higher than that following the oral administration of CIL powder.

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
