# Peer review of "Transdermal System Based on Solid Cilostazol Nanoparticles Attenuates Ischemia/Reperfusion-Induced Brain Injury in Mice"

_nanomaterials, 2021, doi:10.3390/nano11041009_

Round 1

Reviewer 1 Report

This is a potentially interesting manuscript, but several concerns should be resolved before further consideration:
-“ICR mice”-“ICR” should be defined; 
-the authors write “…silicone-coated 8-0 nylon monofilament (Natsume Seisakusyo Co. Ltd., Tokyo, Japan) was inserted into the left middle cerebral artery, which was maintained for 2 h at 37 °C using a heating pad. Then, the middle cerebral artery blood flow was restored by withdrawing the nylon monofilament under anesthesia with isoflurane, before inducing MCAO for 6 h after reperfusion [27]”. 
The sentence should be edited or explained why the authors used a second MCAO “for 6 h after reperfusion”;
-the authors write “Next, the brain was removed at 1:00 p.m. and sliced into three parts (2 mm each), area anterior, area bregma, and area posterior, which were 1 mm anterior and 1 mm posterior to the center of the bregma, using Brain Matrices”. 
The authors should explain it in more detail;
-Vehicle should be described;
-The authors should describe the neurological deficit of mice after stroke and exclusion criteria;
- the authors write “…The infarct volume in the rat brain transdermally treated with the CIL-Ngel was 68% that of the vol-307 ume in the brain orally treated with CIL powder (Figure 6D).”
The authors should describe the rat's stroke experiment or edit the sentence.
-The number of animals should be described for the results of Fig 6 and 7.

Author Response

We carefully revised our manuscript according to the suggestions of the reviewer 1, and details are as follows.

< Q and A for Reviewer 1>

Q1. ICR mice”-“ICR” should be defined.

A1. Thank you very much for pointing this out. In order to respond to the reviewer’s comment, we revised to Institute of Cancer Research (ICR) mice (line 81).

Q2. The authors write “…silicone-coated 8-0 nylon monofilament (Natsume Seisakusyo Co. Ltd., Tokyo, Japan) was inserted into the left middle cerebral artery, which was maintained for 2 h at 37 °C using a heating pad. Then, the middle cerebral artery blood flow was restored by withdrawing the nylon monofilament under anesthesia with isoflurane, before inducing MCAO for 6 h after reperfusion [27]”.

The sentence should be edited or explained why the authors used a second MCAO “for 6 h after reperfusion”.

A2. Thank you for pointing out this. In order to respond to the reviewer’s comment, we collected the sentence (line 190-193).

Q3. The authors write “Next, the brain was removed at 1:00 p.m. and sliced into three parts (2 mm each), area anterior, area bregma, and area posterior, which were 1 mm anterior and 1 mm posterior to the center of the bregma, using Brain Matrices”.

The authors should explain it in more detail.

A3. The reviewer’s comments are very important. Three slices were cut out for analysis: from the bregma (2 mm, areabregma), 1 mm anterior to the bregma (2 mm, areaanterior), and 1 mm posterior to the bregma (2 mm, areaposterior), using Brain Matrices (Brain ScienceIdea Co., Ltd., Osaka, Japan). In order to respond to the reviewer’s comment, we revised the sentence in Materials and Methods (line 195-198).

Q4. Vehicle should be described.

A4. The reviewer’s comment is correct. The 3% carbopol gel containing 0.5% MC, 0.005% docusate sodium salt, 5% HPβCD and 2% L-menthol was used as vehicle. In order to respond to the reviewer’s comment, we described the vehicle in the Materials and Methods (line 118-119).

Q5. The authors should describe the neurological deficit of mice after stroke and exclusion criteria.

A5. Thank you very much for pointing this out. In this study, neurological functional were observed 24 h postoperatively, and the mice with no neurological symptoms or unconscious were excluded. In order to respond to the reviewer’s comment, we added the contents in the Materials and Methods (line 205-206).

Q6. The authors write “…The infarct volume in the rat brain transdermally treated with the CIL-Ngel was 68% that of the vol-307 ume in the brain orally treated with CIL powder (Figure 6D).”

The authors should describe the rat's stroke experiment or edit the sentence.

A6. The reviewer’s comments are very important. In order to respond to the reviewer’s comment, we revised to “Figure 7C” from “Figure 6D” (line 309).

Q7. The number of animals should be described for the results of Fig 6 and 7.

A7. The reviewer’s comment is correct. In order to respond to the reviewer’s comment, we added the number of animals in the Fig. 6 and 7 legends (Fig. 6 and 7 legends).

Thank you for great comments.

Reviewer 2 Report

Development of transdermal system based on solid cilostazol nanoparticles for stroke treatment is very interesting and however, the time is critical for initial stroke treatment. Intravenous administration of drug will be preferred by ER doctors. Transdermal system may serve as a good approach for subsequent dosing delivery as the blood level of CIL was more sustained than that via oral administration of CIL power in this study as the authors stated in the abstract.   

In line 28, it was described that the therapeutic effect of CIL-Ngel was higher than that in the oral absorption of CIL powder.  There is a big dose difference between oral and transdermal administrations. The oral administration dose was 3 mg/kg so it would be 0.075 mg if the body weight of the mouse was 25 grams. However, the transdermal dose was 0.3 g (300 mg). It is not a fair comparison.  The experiment should be performed and compared at same dose and same CIL format.

In line 192, the sentence of “before inducing MCAO for 6 h after reperfusion” is confusing to the readers. MCAO should not be after reperfusion.

In line 195, the brain was sliced into three parts (2 mm each). It is unclear that the area was measured from anterior side or posterior side because only three areas were used for infarct volume measurement. The ideal calculation should be to have the average area for each part of brain measured from both anterior and posterior sides of that part of brain. Please clarify.

In Figure 6D, the CIL concentration in non-ischemia site was higher than that in ischemia site. We expect that the CIL should be well delivered into the ischemia area. Does this mean that CIL was not sufficiently delivered into the ischemia tissue?

In Figure 7A, the images for control, vehicle and CIL are not at same brain level. The vehicle image was at the center of infarct and the CIL image was away from the center.  Please select the images for these groups from similar brain level.  The number of animals was not reported in the legend. It is important to have the sample size reported and also clarify that the mice were randomly assigned into the groups and then infarct area was measured by the person who was blind to the group.

Author Response

We carefully revised our manuscript according to the suggestions of the reviewer 2, and details are as follows.

< Q and A for Reviewer 2>

Q1. In line 28, it was described that the therapeutic effect of CIL-Ngel was higher than that in the oral absorption of CIL powder. There is a big dose difference between oral and transdermal administrations. The oral administration dose was 3 mg/kg so it would be 0.075 mg if the body weight of the mouse was 25 grams. However, the transdermal dose was 0.3 g (300 mg). It is not a fair comparison. The experiment should be performed and compared at same dose and same CIL format.

A1. Thank you very much for pointing this out. The CIL-Ngel contain the 0.5% CIL. Therefore, the dose is 1.5 mg in the mice, and the dose in the transdermal administrations of CIL-Ngel is higher than that in oral administration of CIL powder. On the other hand, the AUC0-66h was similar between of oral and transdermal administrations (Fig. 6C), although, the therapeutic effect of CIL-Ngel was higher than that in the oral absorption of CIL powder. These results showed that the CIL-Ngel was better for the therapy of I/R-induced brain injury in comparison with oral administration of CIL powder under the similar AUC. In order to respond to the reviewer’s comment, we revised the sentence in the Abstract (line 28).

Q2. In line 192, the sentence of “before inducing MCAO for 6 h after reperfusion” is confusing to the readers. MCAO should not be after reperfusion.

A2. The reviewer’s comment is correct. In order to respond to the reviewer’s comment, we collected the sentence (line 190-193).

Q3. In line 195, the brain was sliced into three parts (2 mm each). It is unclear that the area was measured from anterior side or posterior side because only three areas were used for infarct volume measurement. The ideal calculation should be to have the average area for each part of brain measured from both anterior and posterior sides of that part of brain. Please clarify.

A3. Thank you for pointing out this. The infarct area was analyzed with Image J, and the area (mm2) was expressed as the average area for each part of brain measured from both anterior and posterior sides of that part of brain in this study. In order to respond to the reviewer’s comment, we added the contents in Materials and Methods (line 199-201).

Q4. In Figure 6D, the CIL concentration in non-ischemia site was higher than that in ischemia site. We expect that the CIL should be well delivered into the ischemia area. Does this mean that CIL was not sufficiently delivered into the ischemia tissue?

A4. The reviewer’s comments are very important. These results suggested that the blood flow may be limited under the the ischemia, resulting in the delivery of CIL in ischemia site was not lower than that of non-ischemia site. In order to respond to the reviewer’s comment, we added the contents in the discussion (line 382-385).

Q5. In Figure 7A, the images for control, vehicle and CIL are not at same brain level. The vehicle image was at the center of infarct and the CIL image was away from the center.  Please select the images for these groups from similar brain level.

A5. Thank you very much for pointing this out. In order to respond to the reviewer’s comment, we selected the images for these groups from similar brain level (Figure 7A).

Q6. In Figure 7A, the number of animals was not reported in the legend. It is important to have the sample size reported and also clarify that the mice were randomly assigned into the groups and then infarct area was measured by the person who was blind to the group.

A6. The reviewer’s comment is correct. In order to respond to the reviewer’s comment, we added the number of animals in the Fig. 7 legend. In addition, we mentioned for the mice was separated randomly in the method (Figure 7 legend).

Thank you for great comments.

Round 2

Reviewer 1 Report

Before further consideration, the authors should fix the remaining concerns. The authors should clearly describe how long was MCAO and how long was reperfusion in mice. The authors should demonstrate neurological deficit for all animals with MCAO, compare the deficits between animal groups, and describe how many mice were excluded from studies and why (no neurological deficit, mortality rate). The authors should explain why neurological deficit was not measured for all duration of the study but only for 24 h.
line 308-the authors write that "the infarct volume in the rat brain...".
This should be edit or the authors should describe not only the mouse model of MCAO but the rat model too. 

Author Response

We carefully revised our manuscript according to the suggestions of the reviewer 1, and details are as follows.

< Q and A for Reviewer 1>

Q1. The authors should clearly describe how long was MCAO and how long was reperfusion in mice.

A1. Thank you very much for pointing this out. The silicone-coated 8-0 nylon monofilament was inserted into the left middle cerebral artery for 6 h (3:00 p.m.), and housed for 4 h (7:00 p.m.). After that, the CIL was orally (3 mg/kg) or transdermally (0.3 g CIL gel) administered once a day (7:00 p.m.). In order to respond to the reviewer’s comment, we revised the sentence (line 191-199).

Q2. The authors should demonstrate neurological deficit for all animals with MCAO, compare the deficits between animal groups, and describe how many mice were excluded from studies and why (no neurological deficit, mortality rate).

A2. Thank you for pointing out this. The 5 week old ICR mice (41 mice) were operated, and the 8 mice with no neurological symptoms (n=1) or unconscious (n=7) were excluded in this study. The 33 mice with the I/R injury induced by MCAO were separated 5 groups [non-treated group (n=5), group orally treated with vehicle (n=5), group orally treated with CIL powder (n=9), group transdermally treated with vehicle (n=5), group transdermally treated with CIL-Ngel (n=9)]. In order to respond to the reviewer’s comment, we added the information in the collected the sentence in Materials and Methods (line 187-191).

Q3. The authors should explain why neurological deficit was not measured for all duration of the study but only for 24 h.

A3. The reviewer’s comments are very important. In this study, the neurological functional were observed at 4 h, 24 h, 48 h and 63 h after postoperatively. In addition, we confirmed whether the mice were induced the I/R injury via MCAO by the neurological functional 4 h postoperatively. In this study, the 8 mice with no neurological symptoms (n=1) or unconscious (n=7) were excluded, since the mice was died or I/R injury via MCAO was not induced. In order to respond to the reviewer’s comment, we added the contents in Materials and Methods (line 194-198).

Q4. line 308-the authors write that "the infarct volume in the rat brain...".

This should be edit or the authors should describe not only the mouse model of MCAO but the rat model too.

A4. The reviewer’s comment is correct. We collected to mouse brain from rat brain (line 313).

Thank you for great comments.

Reviewer 2 Report

Most of my concerns are resolved. The only question is why the effect was different when AUC in both groups were similar. The effect should be similar as well. The effect of CIL depends on the plasma concentration, not administrative route. The oral dose of CIL is 0.075 mg per mouse and the transdermal dose is 1.5 mg per mouse.  The statement of “These results showed that the CIL-Ngel was better for the therapy of I/R-induced brain injury in comparison with oral administration of CIL powder under the similar AUC” is not a fair comparison. It should be revised. Otherwise, it misleads the readers. If necessary, an additional experiment may be performed using oral administration of CIL-Ngel and traditional CIL power at same dose to detect if CIL-Ngel is better than traditional power.

Author Response

We carefully revised our manuscript according to the suggestions of the reviewer 2, and details are as follows.

< Q and A for Reviewer 2>

Q1. The effect should be similar as well. The effect of CIL depends on the plasma concentration, not administrative route. The oral dose of CIL is 0.075 mg per mouse and the transdermal dose is 1.5 mg per mouse.  The statement of “These results showed that the CIL-Ngel was better for the therapy of I/R-induced brain injury in comparison with oral administration of CIL powder under the similar AUC” is not a fair comparison. It should be revised. Otherwise, it misleads the readers. If necessary, an additional experiment may be performed using oral administration of CIL-Ngel and traditional CIL power at same dose to detect if CIL-Ngel is better than traditional power.

A1. The reviewer’s comments are very important. Although, the AUC0-66h was similar, the drug behavior in blood is different between of oral and transdermal administrations (Fig. 6C). The plasma CIL in mice treated with CIL-Ngel was sustained in comparison with oral administration of CIL power. The difference in drug behavior in blood may relate therapeutic effect for I/R-induced brain injury. On the other hand, further study need to clarify the difference of therapeutic effect between orally and transdermally treated mice. Therefore, we removed the sentence in the Abstract. Thank you very much for pointing this out.

Thank you for great comments.

Round 3

Reviewer 1 Report

It is still some unclear descriptions:
MCAO was 6 hours. How long was reperfusion? 
It should be described not only in days but in hours. 
How long was reperfusion in hours?
How the neurological deficits were calculated? 
What scale was used? 
Is the bigger neurological score means a worse score? It should be described in the method section in detail.
Lines 194-199 "The 5 week old ICR mice (41 mice) were operated, and the 8 mice with no neurological symptoms (n=1) or unconscious (n=7) were excluded in this study. The 33 mice with the I/R injury induced by MCAO were separated 5 groups [non-treated group (n=5), group orally treated with vehicle (n=5), group orally treated with CIL powder (n=9), group transdermally treated with vehicle (n=5), group transdermally treated with CIL-Ngel (n=9)]. In order to respond to the reviewer’s comment, we added the information in the collected the sentence in Materials and Methods (line 187-191)"
Is "n" number of animals or neurological score? If it is a neurological deficit it should be changed for other abbreviation, because it seems like a number of animals. It should be described as, for example, "the neurological deficit was 5 points for the non-treated group (n=6 mice), etc.."

Author Response

We carefully revised our manuscript according to the suggestions of the reviewers, and details are as follows.

< Q and A for Reviewer 1>

Q1. It should be described not only in days but in hours. How long was reperfusion in hours? How the neurological deficits were calculated? What scale was used? Is the bigger neurological score means a worse score? It should be described in the method section in detail.

A1. The reviewer’s comments are very important. The reperfusion was performed for 70 h. The neurological functional was determined on a four-group as follows: 0 = no neurological symptoms; 1 point =inability to completely extend the right front paw; 2 = rotating while crawling and falling to the right side; 3 = inability to walk without assistance; and 4 = unconscious. The mice scoring 0 and 4 were excluded. It was reported that the infarct volume tend to increase with neurological score in the previous studies, although, we don’t have the data on how much correlation in this study. In order to respond to the reviewer’s comment, we added this information in Materials and Methods (line 200-201, 211-214).

Q2. Lines 194-199: Is "n" number of animals or neurological score? If it is a neurological deficit it should be changed for other abbreviation, because it seems like a number of animals. It should be described as, for example, "the neurological deficit was 5 points for the non-treated group (n=6 mice), etc.."

A2. The reviewer’s comment is correct. The "n (number)" is animal number. We showed the neurological deficit as neurological score, and added the score in Materials and Methods. Thank you very much for pointing this out (line 195-198).

Thank you for great comments.
